# Effect of Radiotherapy on the Right Ventricular Function in Lung Cancer Patients

**DOI:** 10.3390/cancers16111979

**Published:** 2024-05-23

**Authors:** Grzegorz Sławiński, Maja Hawryszko, Zofia Lasocka-Koriat, Anna Romanowska, Kamil Myszczyński, Anna Wrona, Ludmiła Daniłowicz-Szymanowicz, Ewa Lewicka

**Affiliations:** 1Department of Cardiology and Electrotherapy, Faculty of Medicine, Medical University of Gdańsk, 80-210 Gdańsk, Poland; gslawinski@gumed.edu.pl (G.S.); ludwik@gumed.edu.pl (L.D.-S.); elew@gumed.edu.pl (E.L.); 21st Department of Cardiology, Faculty of Medicine, Medical University of Gdańsk, 80-210 Gdańsk, Poland; zofia.lasocka@gumed.edu.pl; 3Department of Oncology and Radiotherapy, Faculty of Medicine, Medical University of Gdańsk, 80-210 Gdańsk, Poland; ankapoplawska@gmail.com (A.R.); wronania@gmail.com (A.W.); 4Centre of Biostatistics and Bioinformatics Analysis, Medical University of Gdańsk, 1a Debinki, 80-211 Gdańsk, Poland; kamil.myszczynski@gmail.com

**Keywords:** lung cancer, cardiotoxicity, right ventricle, radiotherapy, speckle tracking echocardiography, 3-dimensional echocardiography

## Abstract

**Simple Summary:**

Due to the location of the right ventricle in the anterior part of the chest, it is particularly vulnerable to damage during radiotherapy, for example, in the treatment of lung cancer. The aim of this prospective study was to compare the effect of radiochemotherapy versus chemotherapy on right ventricular systolic function, assessed using multiple echocardiographic variables. In all patients, echocardiography was performed before the start of treatment, after its completion, and three months after its completion. The analysis of the results included, among others, the radiation dose received by the entire heart and its individual chambers and coronary vessels. The obtained results make it possible to assess the scale of the problem of right ventricular damage after chest radiotherapy and to assess the risk factors associated with the occurrence of this complication.

**Abstract:**

Background: Anticancer treatment is associated with many side effects, including those involving the cardiovascular system. While many studies are available on the effects of radiotherapy (RT) on the left ventricle (LV), studies are lacking on the early effects of RT on the structure and function of the right ventricle (RV). Our study aims to assess, using modern echocardiographic techniques, the effect of irradiation on RV systolic function in the mid-term follow-up of patients undergoing RT for lung cancer (LC). Methods: This single-center, prospective study included consecutive patients with LC who were referred for treatment with definite radiotherapy and chemotherapy (study group) or chemotherapy only (control group). Results: The study included 43 patients with a mean age of 64.9 ± 8.1 years. Cancer treatment-related RV toxicity (CTR-RVT) was found in 17 patients (40%). Early reductions in TAPSE values were observed among patients in the study group (20.3 mm vs. 22.1 mm, *p* = 0.021). Compared to baseline, there was a significant reduction in RV global longitudinal strain (RV GLS) in the study group immediately after the treatment (−21.1% vs. −18.4%, *p* = 0.02) and also at 3 months after RT (−21.1% vs. −19.1%, *p* = 0.021). A significant reduction in the RV FWLS value was also observed at 3 months after the end of the treatment (−23.8% vs. −21.8, *p* = 0.046). There were no significant changes in the three-dimensional right ventricular ejection fraction (3DRVEF) during the follow-up. We found a correlation (*p* = 0.003) between the mean dose of radiation to the RV and 3DRVEF when assessed immediately after RT. The mean dose of radiation to the heart correlated with RV free-wall longitudinal strain (RV FWLS) immediately after RT (*p* = 0.03). Conclusions: RV cardiotoxicity occurs in nearly half of patients treated for lung cancer. TAPSE is an important marker of deterioration of RV function under LC treatment. Compared to 3DRVEF, speckle tracking echocardiography is more useful in revealing deterioration of RV systolic function after radiotherapy.

## 1. Introduction

In recent decades, the survival of cancer patients has improved as a result of earlier diagnosis and the development of new treatment methods. However, lung cancer is still the most common cause of death. In addition, cancer treatment is associated with many side effects, including those involving the cardiovascular system. In patients with locally advanced non-small-cell lung cancer (NSCLC), the radiation dose delivered to the heart was associated with cardiotoxicity and overall survival [1], and also with major adverse cardiac events that occurred relatively early after treatment [2]. While many studies are available on the effects of radiotherapy (RT) on the left ventricle (LV), there has been much less research on the effect of RT on the right ventricle (RV), especially in patients with lung cancer. Our study aims to examine the early effects of RT, within 3 months after the end of the cancer treatment, on the RV structure and function using modern echocardiographic techniques.

## 2. Methods

This prospective single-center study was conducted between November 2020 and March 2022 and included consecutive patients with lung cancer who were referred for treatment with definitive radiotherapy with concurrent chemotherapy (study group) or chemotherapy only (control group). The study was approved by the Independent Bioethics Committee at the Medical University of Gdańsk (NKBBN/575/2020).

The inclusion criteria included the following:-Age over 18;-Histologically proven and unresectable lung cancer;-Planned treatment with radiochemotherapy or chemotherapy;-A written patient consent for the study participation.

The exclusion criteria were as follows: -Planned or previous surgical treatment of lung cancer;-History of potentially cardiotoxic oncological treatment;-No written consent for the study participation.

Transthoracic echocardiography (TTE), including three-dimensional (3D) and speckle tracking echocardiography (STE) was performed at baseline (before the initiation of cancer treatment), in the first week after the end of RT (after 4 cycles of chemotherapy in the control group), and 12 weeks after the end of the treatment. Medical history and treatment data were collected from the electronic medical charts. Hypercholesterolemia was defined as low-density lipoprotein (LDL) cholesterol >130 mg/dL or statin use, and the patients were considered to have chronic kidney disease when the estimated glomerular filtration rate (eGFR) was <60 mL/min/1.73 m^2^ in the last two assessments. Echocardiography was performed at each stage using the same protocol, personnel, and equipment (Vivid S95 system, General Electric Medical Health). Examinations were performed using parasternal, apical, and subcostal views, including dedicated RV views. Cine loops from three standard apical views (4-, 3-, and 2-chamber) were recorded for off-line analysis (EchoPac 201, GE). Measurements of RV dimensions and the assessment of RV systolic function were performed according to the ASE recommendations [3]. Following the analysis of the limitations of each parameter (Appendix A), the RV systolic function was assessed using the following parameters: tricuspid annular plane systolic excursion (TAPSE), tissue Doppler-derived tricuspid annulus systolic velocity (RV S’), RV global longitudinal strain (RV GLS), RV free-wall longitudinal strain (RV FWLS), and 3D RV ejection fraction (3DRVEF).

Asymptomatic cancer treatment-related RV toxicity (CTR-RVT) was defined as follows:-New 3DRVEF reduction by >10% or new 3DRVEF reduction by >5 percentage points to a 3DRVEF <45% [4]; or-New relative reduction in either RV GLS or RV FWLS by ≥10% [5].

Asymptomatic cancer treatment-related LV toxicity (CTR-LVT) was defined according to the recent European Society of Cardiology guidelines on cardio-oncology [6]:-New LV ejection fraction (LVEF) reduction to <40% indicates severe LV cardiotoxicity;-New LVEF reduction by ≥10% to a LVEF of 40–49% or new LVEF reduction by <10% to a LVEF of 40–49% and either new relative decline in LV GLS by >15% from baseline or new rise in cardiac biomarkers indicate moderate LV cardiotoxicity;-LVEF ≥ 50% and a new relative decline in GLS by >15% from baseline and/or a new rise in cardiac biomarkers indicate mild LV cardiotoxicity.

RT planning and treatment were conducted in accordance with the local institutional protocol. The planning target volume (PTV) dose was mostly 66 Gy delivered in 30 daily fractions. In most cases, 4-dimensional scans (4D CT) were used for planning purposes and RT was delivered with dynamic techniques. All patients were irradiated with 6 MeV photon beams in a linear accelerator (TrueBeam^®^ SN1403 accelerator, Varian Medical Systems Inc., Palo Alto, CA, USA). Individual structures of the heart including the left atrium (LA), right atrium (RA), RV, LV, left anterior descending coronary artery (LAD), left circumflex coronary artery (Cx), and right coronary artery (RCA) were contoured on an ‘average’ set of 4D CT scans or a non-contrast enhancement scan of conventional planning CT based on the existing atlases by two radiation oncologists [7]. Dosimetric analysis was performed using dose distribution data of Eclipse Radiotherapy Planning System (Varian Medical Systems Inc., Palo Alto, CA, USA). Dose-volume histograms were generated for organs at risk. Parameters for analysis included the mean heart dose, mean pericardial dose, mean dose for RV, LV, RA, LA, LAD, Cx, and RCA, heart volume receiving a radiation dose of 5 Gy (V5 Gy) and 30 Gy (V30 Gy), and RV, LV, RA, and LA volume receiving a dose of 5 Gy and 30 Gy.

## 3. Statistical Analysis

Continuous variables were expressed as the mean ± SD if normally distributed or median if not normally distributed. In the case of continuous variables, normal distribution was tested using the Shapiro-Wilk test. Categorical data were expressed as numbers and percentages. Continuous variables were compared using the independent-sample parametric (unpaired Student *t*) or nonparametric (Mann–Whitney *U*, Kruskal–Wallis test by ranks) tests. Categorical variables were compared using the chi-square test or the Fisher exact test when appropriate. Longitudinal data were compared using a paired *t*-test. Statistical significance values were corrected for multiple comparisons using the Benjamini–Hochberg procedure. Correlations between study group parameters and RT parameters were assessed using Spearman’s rank correlation. Data were analyzed with the use of an R 4.3.2 (R Core Team, 2021) environment. The *p* value of <0.05 was considered statistically significant.

## 4. Results

The study included 43 patients with a mean age of 64.9 ± 8.1 years, including 25 men (58.1%). There were 23 patients in the study group and 20 in the control group. The three planned follow-up visits were completed by 60% of patients. During the follow-up, eight patients (19%) withdrew from the study because of their poor general condition and nine patients (21%) died. Eight patients died due to cancer, and one patient died in a traffic accident.

The baseline characteristics of the studied population are shown in Table 1. The most common NSCLC was adenocarcinoma (46.5%) and squamous cell carcinoma (46.5%). The most common comorbidities were hypertension, diabetes, and hypercholesterolemia, and 91% of the patients were active or former smokers. Chemotherapeutic agents used included cisplatin (*n* = 19), pemetrexed (*n* = 17), and carboplatin (*n* = 15). There were no differences in the chemotherapy regimen between the groups.

The median dose of radiation to the whole heart was 9.4 Gy, with no significant differences between patients with left-sided and right-sided lung cancer. No patient received a mean heart dose of >25 Gy, which, according to the latest guidelines, indicates a very high risk of RT-related cardiotoxicity [6]. In the study group, 21.74% of patients received a mean heart dose of 15–25 Gy, which indicates a high risk of cardiotoxicity, 56.52% received a dose of 5–15 Gy, which is associated with a moderate risk of cardiotoxicity, and 21.74% received a dose of <5 Gy, indicating a low risk of cardiotoxicity [6]. When comparing patients with left-sided and right-sided lung cancer, there were no differences in the mean radiation doses to the RV. In contrast, patients with left-sided lung cancer received significantly higher doses of LAD and Cx, and patients with right-sided lung cancer received higher doses of RA (Table 2).

Data on RV systolic function were obtained in almost all echocardiographic examinations, and the measurement feasibility was 97.7% for TAPSE, 95.3% for RV S’, 90.7% for RV FWLS, 90.7% for RV GLS, and 88.4% for 3DRVEF. Among the parameters reflecting the RV systolic function (Table 3), compared to baseline, a significant reduction in RV GLS immediately after RT (−21.1% vs. −18.4%, *p* = 0.001) and at 3 months after RT (−21.1% vs. −19.1%, *p* = 0.016) was found in the study group. Early reductions in TAPSE values were also observed among patients in the study group (20.3 mm vs. 22.1 mm, *p* = 0.021). In addition to a significant reduction in the RV GLS value, a significant reduction in the RV FWLS value was also observed in the study group at 3 months after the end of the treatment (−23.8% vs. −21.8, *p* = 0.046). There were no significant changes in 3DRVEF during follow-up.

CTR-RVT was found in seventeen patients (40%), including ten patients in the study group and seven patients in the control group. The diagnosis was based on a decrease in RV longitudinal strain in twelve patients, while a decrease in 3DRVEF was noted in seven patients, as found in an examination performed three months after completion of treatment compared to baseline. Importantly, only two patients met both criteria for the diagnosis of CTR-RVT. In patients with CTR-RVT; however, no significant changes in TAPSE and RV S’ were found. Examples of GLS RV, RV FWLS, and 3DRVEF changes are shown in Figure 1.

CTR-LVT was diagnosed in nine patients (21%), including six patients in the study group (23%) and three patients in the control group (15%), and was categorized as mild in six patients (14%), moderate in one patient (2%), and severe in two patients (5%). In patients with heart failure (HF) at baseline, we noted neither deterioration of LV systolic function after oncological treatment nor HF exacerbations requiring hospitalization. Four patients were diagnosed with both LV and RV cardiotoxicity.

We found a correlation (*p* = 0.003) between the mean RV radiation dose and 3DRVEF when assessed immediately after RT (Figure 2). Although there was no statistically significant reduction in 3DRVEF at this time point compared to baseline, a trend towards such a reduction could be noted (49.7% vs. 52.3%, *p* = 0.08). The mean heart dose correlated with the RV FWLS immediately after RT (*p* = 0.03). Such associations were not found for measurements made 3 months after RT, as well as for other RV parameters.

When evaluating LV function, echocardiography performed 3 months after the completion of RT showed a significant reduction in LV GLS (−17.2% vs. −16.5%, *p* = 0.03) without a significant change in 3DLVEF (Appendix A). There was a correlation between the dose applied to LAD and LV GLS in echocardiography performed 3 months after RT (*p* = 0.04) (Figure 3).

## 5. Discussion

Our study is the first to assess the RV systolic function in lung cancer patients in the early period after definite chemoradiotherapy in comparison with chemotherapy alone using modern echocardiographic techniques (3D and STE). We focused on the RV also in relation to the radiation dose applied to the heart. The results of the present study may inform our understanding of the natural history of RV damage after RT. We found that the adverse effects of RT on RV may already be noted in the first 3 months after completion of RT. This was reflected by the reduction of RV GLS and RV FWLS, and was evident early–immediately after the end of RT. Importantly, TAPSE, one of the standard parameters for the assessment of RV systolic function, also reflected an early deterioration of RV systolic function. Our findings indicate that CTR-RVT occurs much earlier than reported by Chen et al. in patients with stage III NSCLC, who demonstrated RV longitudinal strain worsening at 6 months after concomitant chemoradiotherapy, as indicated by a decrease in RV GLS and RV FWLS [8].

At the same time, we found that the population of lung cancer patients has some specific characteristics and is very different from the populations in which most studies on RT cardiotoxicity have been conducted, for instance, breast cancer or Hodgkin lymphoma survivors. In addition to differences in chest irradiation alone, lung cancer patients are older at the time of treatment and have more cardiac risk factors. They are also characterized by much worse imaging conditions, as most of them are previous or current smokers (91% of patients in our study) and have COPD (69% in our group). Lung cancer and the applied RT also have an impact. In our opinion, all this makes it difficult to compare the above-mentioned populations with lung cancer patients.

Patients in the chemotherapy-only group had significantly higher Charlson Comorbidity Index values compared to the chemoradiotherapy group. The difference is most likely due to the advancement of the cancer and qualification for the appropriate type of treatment. When qualifying for oncological treatment, in addition to assessing the advancement of the cancer, comorbidity is also taken into account. Patients from the study group, being qualified for chemoradiotherapy, due to the fact that this treatment is more burdensome, must be in a relatively good functional condition, and not burdened with multi-morbidities (low Charlson Comorbidity Index), while patients from the control group, qualified only for chemotherapy (often palliative) received this treatment, of course, due to more advanced cancer disease and also due to multi-morbidities (high Charlson Comorbidity Index).

An important part of our study was the assessment of dosimetric parameters in relation to particular heart structures, also in relation to the location of the tumor (left or right). The median heart dose in our group was 9.4 Gy, and in 56% of patients, it was between 5 and 15 Gy which, according to the recent 2022 ESC guidelines on cardio-oncology, indicates a moderate risk of RT-related cardiotoxicity [6]. The mean RV and LV doses were relatively low (<5 Gy), similar to those reported by Chen et al. [8]. In contrast, patients with right-sided lung cancer received a significantly higher dose for RA (mean dose 14.98 ± 13.46 Gy), more than double that of patients with left-sided lung cancer. Patients with left-sided lung cancer received particularly high doses for LAD and Cx (17.7 ± 13.8 Gy for Cx). There was a trend for reduction in 3DRVEF (49.7% vs. 52.3%, *p* = 0.08), and a correlation was found (*p* = 0.003) between the mean RV radiation dose and 3DRVEF assessed immediately after RT. The mean heart dose correlated with the RV FWLS immediately after RT (*p* = 0.03). In addition, a significant correlation was found between the mean dose applied to LAD and LV GLS. When analyzing the radiation doses applied to particular heart structures, it seems that the RV is less exposed to the cardiotoxic effects of radical lung cancer radiotherapy compared to the LV, and the same applies to the RA compared to the LA. Nevertheless, CTR-RVT occurred more frequently than CTR-LVT. This may be due to a thinner RV muscle, which is more susceptible to damage compared to that of the LV, and to the blood supply mainly through one major coronary artery, so there is a lower chance of adequate collateral circulation in case of radiation damage.

Due to the RV location (behind the sternum) and unique shape, the assessment of RV systolic function is often challenging. Unlike LV ejection fraction, there is no single parameter that is reliable, widely available, and easy to measure when assessing RV systolic function [9]. Importantly, those obtained from 2D echocardiography do not reflect the function of the entire RV and this technique does not allow precise measurement of RV volumes and RVEF [10]. In our study, we used various parameters for RV function assessment, being aware of their limitations (as summarized in Appendix A).

Initially, RV was neglected in studies on the cardiotoxicity of oncological treatment. However, the number of reports on this issue has increased in recent years. Interestingly, early RV dysfunction (reduction of RV GLS) paralleled LV involvement and predicted subsequent LV subclinical dysfunction, as demonstrated in breast cancer patients treated with anthracycline, trastuzumab, and paclitaxel [11]. Laufer-Perl et al. reported that CTR-RVT may be even more frequent than CTR-LVT. In a study of 40 breast cancer patients, echocardiography was performed before and after chemotherapy with anthracycline. They reported that 75% and 58% of patients showed a ≥10% relative reduction in RV GLS and RV FWLS, respectively. Interestingly, LVEF and LV GLS remained within normal limits [5].

As already mentioned, we also noted a higher incidence of CTR-RVT than CTR-LVT (40% vs. 21% of patients). A relatively high percentage of CTR-RVT also in patients in the control group (35%) is noteworthy, probably resulting from comorbidities (e.g., hypertension in 60% of patients) and multiple cardiovascular risk factors.

In recent years, there has been a growing interest in strain evaluation in the diagnosis of CTR-RVT and it has been postulated that the RV GLS and RV FWLS may be useful for this purpose. In our RT-treated patients, we found a decrease in GLS RV (immediately and 3 months after RT compared to baseline) and GLS RV FWLS (3 months after RT compared to baseline). In our opinion, taking into account the technical difficulties in performing echocardiography in lung cancer patients and the limited availability of 3D echocardiography, it seems that both these parameters and TAPSE are the most important for monitoring RV function in this group of patients.

However, the cut-off for a definite diagnosis of CTR-RVT still remains unclear [8,12,13,14,15,16,17]. In a study of breast cancer patients, Keramida et al. postulated a relative decline in RV GLS by >14.8%, while Wang et al. suggested RV GLS ≥18.2% for the recognition of RV injury [14,18]. Importantly, strain assessment not only allows the diagnosis of CTR-RVT, but also provides direct data regarding the prognosis. In patients treated with chemoradiotherapy for stage III NSCLC, both baseline RV FWLS and a relative decrease in RV FWLS have been shown to be independent predictors of all-cause mortality, with a worse prognosis in patients with a ≥10.1% reduction in RV FWLS [8].

Some authors consider the 3D STE echocardiography to be the most sensitive method for detecting subclinical RV dysfunction when the conventional parameters reflecting RV function still remain unchanged [19]. In our study, 2D STE was more sensitive than 3DRVEF for the diagnosis of CTR-RVT (71% vs. 41% of patients with CTR-RVT, respectively). This may be surprising given the current strong position of 3D echocardiography in the assessment of RV function. However, this may be related to the specific characteristics of patients with lung cancer, in whom imaging conditions are difficult, especially in comparison to patients with breast cancer or lymphoma. Nevertheless, due to the limited availability of 3D echocardiography, the demonstrated usefulness of 2D STE in patients with lung cancer is of important practical significance.

It was documented that cardiac radiation dose was a predictor of major adverse events and all-cause mortality in patients with locally advanced NSCLC within 2 years after RT [2]. Mean heart dose ≥10 Gy vs. <10 Gy indicated significantly worse overall survival in patients without documented CAD. Importantly, RV dysfunction as a result of oncological treatment adversely affects the prognosis. This has been documented by Shen et al. who showed that the relative reduction of RVEF was the strongest risk factor for cardiovascular events in lymphoma patients treated with anthracycline-containing chemotherapy [20]. In our study, however, it was difficult to assess such relationships due to the short duration of the study.

## 6. Limitations of the Study

The study sample was small, and only 60% of patients underwent the planned three examinations (nine patients died). We did not perform longitudinal deformation analysis in individual RV segments. Data on the radiation technique were not available. The drug therapy used in the two groups was not compared. Follow-up duration was too short to assess the effect of CTR-RVT on the prognosis in lung cancer patients.

## 7. Conclusions

Although the RV is less exposed than the LV during RT for lung cancer, CTR-RVT occurs in almost half of patients in the first 3 months after RT, more frequently than CTR-LVT. The RV longitudinal strain impairment may be noted immediately after the end of RT and persists after 3 months. TAPSE is very useful for monitoring RV function in patients treated for lung cancer. In contrast to modern echocardiographic techniques (RV GLS, 3DRVEF), it is easy to measure and widely available. Measurement of TAPSE should be included in the routine echocardiographic assessment in patients with lung cancer treated with radiotherapy. When planning RT, the mean radiation dose to the heart, as well as the dose to the base of the heart (where the large vessels are located) should be reduced whenever possible to reduce the risk of CRT-RVT.

## Figures and Tables

**Figure 1 cancers-16-01979-f001:**
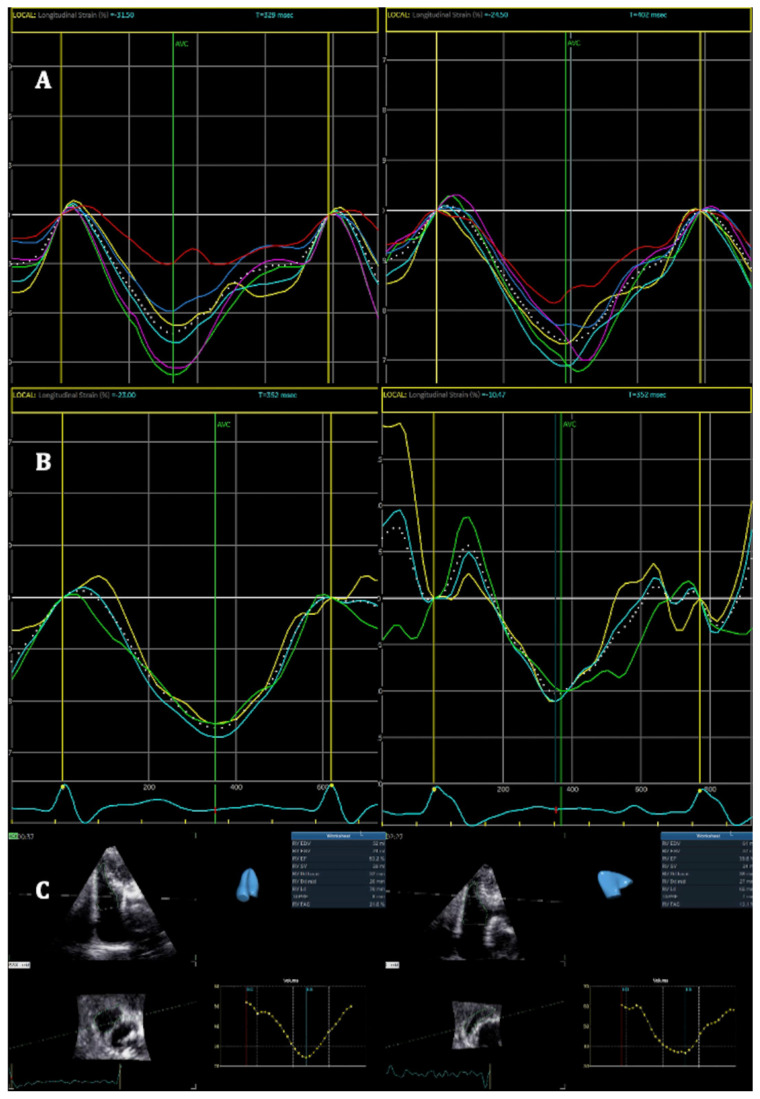
Examples of radiotherapy-induced right ventricular (RV) cardiotoxicity in patients from the study group. (**A**) Changes in right ventricular global longitudinal strain before the start of oncological treatment (**left side**) and 3 months after its completion (**right side**). (**B**) Changes in right ventricular free-wall longitudinal strain before the start of oncological treatment (**left side**) and 3 months after its completion (**right side**). (**C**) Changes in 3D right ventricular ejection fraction before the start of oncological treatment (**left side**) and 3 months after its completion (**right side**).

**Figure 2 cancers-16-01979-f002:**
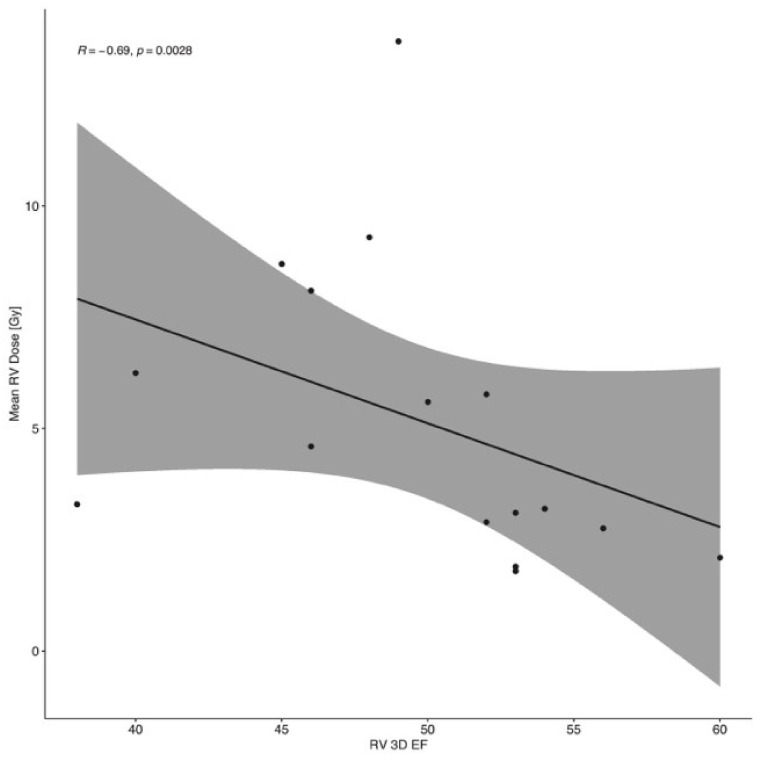
Correlation between the mean RV dose and 3DRVEF.

**Figure 3 cancers-16-01979-f003:**
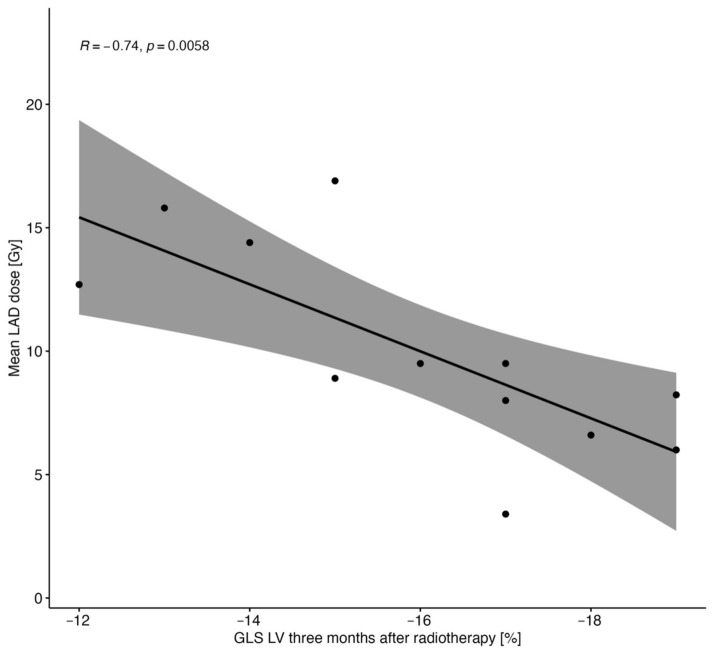
Correlation between LAD radiation dose and LV GLS.

**Table 1 cancers-16-01979-t001:** Baseline characteristics of the study population.

Variable	All Patients (*n* = 43)	Study Group (*n* = 23)	Control Group (*n* = 20)	*p* Value
Age, years	64.9 ± 8.1	64.4 ± 8.6	65.5 ± 7.5	0.68
Males, *n* (%)	25 (58.1%)	13 (56.5%)	12 (60.0%)	0.82
Comorbidities, *n* (%):				
Hypertension	26 (60.5%)	14 (60.1%)	12 (60.0%)	1.00
Coronary artery disease	7 (16.3%)	3 (13.0%)	4 (20.0%)	0.69
Heart failure	6 (13.9%)	3 (13.0%)	3 (15.0%)	0.32
Atrial fibrillation	14 (32.6%)	6 (26.1%)	8 (40.0%)	0.10
Hypercholesterolemia	7 (16.3%)	6 (26.1%)	1 (5.0%)	0.52
Diabetes	1 (2.3%)	1 (4.3%)	0	0.10
Chronic kidney disease	5 (11.6%)	2 (8.7%)	3 (15.0%)	1.00
COPD	5 (11.6%)	2 (8.7%)	3 (15.0%)	1.00
Charlson Comorbidity Index, median	2 (0–6)	1 (0–3)	6 (2–7)	**0.029**
Smoking history, *n* (%):				1.00
Current	12 (27.9%)	6 (26.1%)	6 (30%)	
Former	27 (62.8%)	15 (65.2%)	12 (60%)	
Never	4 (9.3%)	2 (8.7%)	2 (10%)	
Number of pack-years, *n*	37.5 ± 20.2	38.5 ± 20.6	36.3 ± 20.2	0.73
Histological type of lung cancer, *n* (%):				0.077
Adenocarcinoma	20 (46.5%)	9 (39.1%)	11 (55.0%)	
Squamous cell carcinoma	20 (46.5%)	14 (60.1%)	6 (30.0%)	
Small cell lung cancer	2 (4.7%)	0	2 (10.0%)	
Carcinoma not otherwise specified	1 (2.3%)	0	1 (5.0%)	
Death, *n* (%):	9 (20.9%)	5 (21.7%)	4 (20%)	1.00

Abbreviations: COPD—chronic obstructive pulmonary disease.

**Table 2 cancers-16-01979-t002:** Characteristics of radiation treatment depending on the location of lung cancer.

	All Patients (*n* = 23)	Left-Sided Lung Cancer (*n* = 11)	Right-Sided Lung Cancer (*n* = 12)	*p* Value
PTV, mL	610.94 ± 357.67	617.99 ± 422.66	601.77 ± 272.69	0.83
PTV-heart, mL	11.74 ± 12.60	12.04 ± 13.64	11.35 ± 11.82	0.88
Mean heart dose, Gy	9.71 ± 5.43	9.65 ± 5.81	9.79 ± 5.19	0.95
Heart V5 Gy, %	40.46 ± 23.22	39.22 ± 24.26	42.06 ± 22.98	0.78
Heart V30 Gy, %	9.47 ± 7.72	9.58 ± 8.26	9.33 ± 7.39	0.94
Mean pericardium dose, Gy	17.75 ± 5.81	18.65 ± 6.72	16.58 ± 4.42	0.38
Mean RV dose, Gy	4.38 ± 3.18	4.19 ± 3.78	4.61 ± 2.36	0.75
RV V5 Gy, %	29.77 ± 27.17	28.53 ± 31.77	31.37 ± 21.29	0.44
RV V30 Gy, %	0.22 ± 0.77	0.05 ± 0.14	0.45 ± 1.14	0.67
Mean LV dose, Gy	4.61 ± 4.14	5.68 ± 4.88	3.22 ± 2.50	0.13
LV V5 Gy, %	23.79 ± 25.50	25.59 ± 23.97	21.44 ± 28.50	0.53
LV V30 Gy, %	1.91 ± 4.91	3.32 ± 6.25	0.07 ± 0.14	0.061
Mean RA dose, Gy	9.93 ± 10.01	6.05 ± 5.45	14.98 ± 12.46	**0.032**
RA V5 Gy, %	39.33 ± 31.02	29.42 ± 29.99	42.30 ± 27.51	0.11
RA V30 Gy, %	10.45 ± 18.66	3.05 ± 6.16	20.08 ± 24.87	**0.005**
Mean LA dose, Gy	17.45 ± 10.36	18.05 ± 11.78	16.67 ± 8.72	0.75
LA V5 Gy, %	64.45 ± 30.05	60.41 ± 34.45	59.79 ± 29.73	0.45
LA V30 Gy, %	21.26 ± 19.28	24.6 ± 22.36	16.92 ± 14.31	0.33
Mean LAD dose, Gy	9.6 ± 5.39	11.63 ± 6.18	6.97 ± 2.53	**0.025**
Mean Cx dose, Gy	12.56 ± 12.01	17.7 ± 13.77	5.87 ± 3.42	**0.010**
Mean RCA dose, Gy	6.01 ± 7.28	4.51 ± 3.88	7.97 ± 10.10	0.37

Abbreviations: Cx—left circumflex coronary artery; LA—left atrium; LAD—left anterior descending coronary artery; LV—left ventricle; PTV—planning target volume; RA—right atrium; RCA—right coronary artery; RV—right ventricle; V5 Gy—heart volume receiving radiation dose of 5 Gy; V30 Gy—heart volume receiving radiation dose of 30 Gy.

**Table 3 cancers-16-01979-t003:** Changes in right ventricular (RV) dimensions and parameters reflecting the RV systolic function.

All Patients (*n* = 43)					
Variable	Baseline	Immediately after Treatment	*p* Value	Three Months after Treatment	*p* Value (vs. Baseline)
3DRVEF, %	52.0 ± 7.0	51.1 ± 6.6	0.34	50.4 ± 6.0	0.56
RV FWLS, %	−22.5 ± 5.6	−23.2 ± 5.2	0.60	−22.2 ± 5.9	0.27
RV GLS, %	−20.0 ± 5.3	−19.0 ± 3.6	0.09	−20.6 ± 4.5	0.34
RV S’, cm/s	13.7 ± 3.3	13.4 ± 2.2	0.56	13.7 ± 2.8	0.79
TAPSE, mm	22.4 ± 3.4	21.5 ± 3.9	0.30	21.1 ± 3.4	0.09
RVOT, mm	32.9 ± 4.6	32.5 ± 3.7	0.19	33.3 ± 3.9	0.88
RVIT, mm	33.4 ± 3.4	33.4 ± 3.4	0.91	34.5 ± 4.4	0.36
RA area, cm^2^	14.1 ± 3.3	14.4 ± 2.8	0.43	14.5 ± 3.4	0.75
Study group (*n* = 23)					
3DRVEF, %	52.3 ± 6.3	49.7 ± 5.7	0.08	49.4 ± 4.2	0.40
RV FWLS, %	−23.8 ± 4.5	−22.9 ± 5.3	0.38	−21.8 ± 5.8	**0.046**
RV GLS, %	−21.1 ± 4.0	−18.4 ± 4.1	**0.001**	−19.1 ± 4.3	**0.016**
RV S’, cm/s	13.5 ± 3.5	12.4 ± 1.9	0.41	12.8 ± 2.3	0.91
TAPSE, mm	22.1 ± 3.3	20.3 ± 3.2	**0.021**	20.8 ± 3.3	0.051
RVOT, mm	33.2 ± 5.4	32.9 ± 2.2	0.07	33.1 ± 3.4	0.09
RVIT, mm	34.3 ± 3.1	33.7 ± 3.3	0.24	36.0 ± 3.2	0.18
RA area, cm^2^	15.2 ± 4.2	14.4 ± 3.3	0.37	16.2 ± 3.7	0.60
Control group (*n* = 20)					
3DRVEF, %	51.8 ± 7.9	52.5 ± 7.2	0.73	51.7 ± 7.9	0.92
RV FWLS, %	−21.2 ± 6.4	−23.6 ± 5.2	0.22	−22.8 ± 6.4	0.82
RV GLS, %	−18.8 ± 6.2	−19.8 ± 2.9	0.80	−22.8 ± 4.1	0.36
RV S’, cm/s	14.0 ± 3.0	14.5 ± 2.0	0.81	14.8 ± 3.1	0.81
TAPSE, mm	22.8 ± 3.7	22.8 ± 4.2	0.31	21.6 ± 3.6	0.55
RVOT, mm	32.6 ± 3.6	32.2 ± 4.8	0.14	33.5 ± 4.6	0.05
RVIT, mm	32.5 ± 3.5	32.9 ± 3.6	0.94	32.7 ± 5.0	0.95
RA area, cm^2^	13.0 ± 1.5	14.3 ± 2.2	0.03	12.5 ± 1.5	0.74

Abbreviations: 3DRVEF—three-dimensional RV ejection fraction; RV FWLS—RV free wall longitudinal strain; RA—right atrium; RV S’—tissue Doppler-derived tricuspid annulus systolic velocity; RVIT—RV inflow tract; RVOT—RV outflow tract; TAPSE—tricuspid annular plane systolic excursion.

## Data Availability

The datasets used and/or analyzed during the current study are available from the corresponding author upon reasonable request.

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
