# Peer review of "Effect of Radiotherapy on the Right Ventricular Function in Lung Cancer Patients"

_cancers, 2024, doi:10.3390/cancers16111979_

Round 1
Reviewer 1 Report
Comments and Suggestions for Authors
In their article "Effect of radiotherapy on the right ventricular function in lung cancer patients" Sławiński et al. presented their interesting and novel data. As has been demonstrated the chosen echocardiographic parameters correspond with early subclinical cardiotoxicity induced by radiotherapy. The article is well-written. I have only some minor comments:
1. Abstract section. In conclusions you mentioned the differences in TAPSE which were not listed. Please correct.
2. Please correct the numbering of supplementary tables in the order of their appearance in the text.
3. "The three planned follow-up visits were completed by 60% of patients." It is the main study limitation. However, it is normal in that type of analyses in that patients.
4. The groups were significantly different in Charlson CI. Please explain that.
5. Figure 1. Please explain the introduced markings.
6. What about multivariable analyses? I would suggest the one focused on RV GLS or 3DRVEF.
Author Response
In their article "Effect of radiotherapy on the right ventricular function in lung cancer patients" Sławiński et al. presented their interesting and novel data. As has been demonstrated the chosen echocardiographic parameters correspond with early subclinical cardiotoxicity induced by radiotherapy. The article is well-written. I have only some minor comments:
- Abstract section. In conclusions you mentioned the differences in TAPSE which were not listed. Please correct.
Authors' response: Corrected. Information added: "Early reductions in TAPSE values ​​were observed among patients in the study group (20.3 mm vs 22.1 mm, p=0.021)."
- Please correct the numbering of supplementary tables in the order of their appearance in the text.
Authors' response: Corrected.
- "The three planned follow-up visits were completed by 60% of patients." It is the main study limitation. However, it is normal in that type of analyses in that patients.
Authors' response: Thank you for this comment. We agree that this is one of the main limitations (as reported in the relevant section). Unfortunately, this is due to the specific nature of the cardio-oncological population, which has a higher mortality rate and significantly impaired functional status, which often resulted in the inability to make it to a follow-up appointment.
- The groups were significantly different in Charlson CI. Please explain that.
Authors' response: Thank you for this comment. The difference in CCI is most likely due to the advancement of the cancer and qualification for the appropriate type of treatment. When qualifying for oncological treatment, in addition to assessing the advancement of the cancer, co-morbidity is also taken into account. Patients from the study group, being qualified for chemoradiotherapy, due to the fact that this treatment is more burdensome, must be in a relatively good functional condition, not burdened with multi-morbidities (low CCI), while patients from the control group, qualified only for chemotherapy (often palliative) received this treatment, of course, due to a more advanced cancer disease and multi-morbidities (high CCI). We have added appropriate information in the manuscript.
- Figure 1. Please explain the introduced markings.
Authors' response: Corrected.
- What about multivariable analyses? I would suggest the one focused on RV GLS or 3DRVEF.
Authors' response: Thank you for this comment. Of course, we tried to evaluate potential predictors of RV cardiotoxicity using multivariable analyses. However, in no model did we obtain statistically significant data confirming the dominant contribution of any factor (mainly in the context of radiotherapy variables) in the development of RV cardiotoxicity. This may be due primarily to the small (from a statistical point of view) size of the study group.
Reviewer 2 Report
Comments and Suggestions for Authors
Dear authors,
This study is necessary, valid, and possibly can begin a series of new projects dealing with the important issue of radiotherapy in the population with lung cancer.
Although the studied population was small, as pointed out by authors, the echocardiographic methodology and ecquipment was adequate. The statistical methods utilized were correct and the presented results and comments including limitations seem also adequate.
The reviewer would like to ask a question regarding the following statement: "Although there was no statistically significant reduction in 3DRVEF at this time point compared to baseline, a trend towards such a reduction could be noted (49.7% vs 52.3%, p=0.08).": do the authors think that a larger patient population's data could surpass this somehow bad result, in the reviewers, viepoint?
Yours,
Comments on the Quality of English Language
There are only the need of minor English revisions and in the general its use was of good quality.
Author Response
Dear authors,
This study is necessary, valid, and possibly can begin a series of new projects dealing with the important issue of radiotherapy in the population with lung cancer.
Although the studied population was small, as pointed out by authors, the echocardiographic methodology and ecquipment was adequate. The statistical methods utilized were correct and the presented results and comments including limitations seem also adequate.
The reviewer would like to ask a question regarding the following statement: "Although there was no statistically significant reduction in 3DRVEF at this time point compared to baseline, a trend towards such a reduction could be noted (49.7% vs 52.3%, p=0.08).": do the authors think that a larger patient population's data could surpass this somehow bad result, in the reviewers, viepoint?
Yours,
Authors’ response: Thank you for this comment. We agree with the statement made by the reviewer - one of the reasons that did not make it possible to achieve statistical significance in this case is the relatively small number of patients (from a statistical point of view). The second reason is probably more difficult 3D imaging in this group of patients, which we also presented in the results section (we were able to assess RV GLS and RV FWLS in more patients than 3D RVEF).